# *Cannabis sativa* L. as a Natural Drug Meeting the Criteria of a Multitarget Approach to Treatment

**DOI:** 10.3390/ijms22020778

**Published:** 2021-01-14

**Authors:** Anna Stasiłowicz, Anna Tomala, Irma Podolak, Judyta Cielecka-Piontek

**Affiliations:** 1Department of Pharmacognosy, Poznan University of Medical Sciences, Swiecickiego 4, 61-781 Poznan, Poland; astasilowicz@ump.edu.pl; 2Department of Pharmacognosy, Medical College, Jagiellonian University, Medyczna 9, 30-688 Cracow, Poland; annatomala95@gmail.com (A.T.); irma.podolak@uj.edu.pl (I.P.)

**Keywords:** *Cannabis*, phytocannabinoids (THC and CBD), terpenes, multitarget, receptors

## Abstract

*Cannabis sativa* L. turned out to be a valuable source of chemical compounds of various structures, showing pharmacological activity. The most important groups of compounds include phytocannabinoids and terpenes. The pharmacological activity of *Cannabis* (in epilepsy, sclerosis multiplex (SM), vomiting and nausea, pain, appetite loss, inflammatory bowel diseases (IBDs), Parkinson’s disease, Tourette’s syndrome, schizophrenia, glaucoma, and coronavirus disease 2019 (COVID-19)), which has been proven so far, results from the affinity of these compounds predominantly for the receptors of the endocannabinoid system (the cannabinoid receptor type 1 (CB_1_), type two (CB_2_), and the G protein-coupled receptor 55 (GPR_55_)) but, also, for peroxisome proliferator-activated receptor (PPAR), glycine receptors, serotonin receptors (5-HT), transient receptor potential channels (TRP), and GPR, opioid receptors. The synergism of action of phytochemicals present in *Cannabis* sp. raw material is also expressed in their increased bioavailability and penetration through the blood–brain barrier. This review provides an overview of phytochemistry and pharmacology of compounds present in *Cannabis* extracts in the context of the current knowledge about their synergistic actions and the implications of clinical use in the treatment of selected diseases.

## 1. Introduction

The history of many drugs currently used in medicine has its origins in plant raw materials. Morphine, atropine, paclitaxel, and, recently, artemisinin are the most famous examples in the drug discovery process [1]. With the development of analytical techniques and separation methods, it was possible to isolate individual compounds and, consequently, define their molecular mechanisms of action by assessing their interactions with selected receptors. Often, the therapeutic demand for compounds of plant origin is enormous, and effective synthesis pathways have to be developed, as in the case of morphine [2]. It took, however, many years from its discovery to a chemical synthesis. At the beginning of the nineteenth century, Friedrich Sertürner, for the first time in history, isolated this alkaloid from the opium poppy (*Papaver somniferum*) [3]. However, it was not until over 100 years later that Sir Robert Robinson first proposed the correct morphine structure in 1925, which was confirmed by the first total synthesis of morphine by Gates and Tschudi in 1952 [4,5,6]. Since then, many attempts have been made to develop extraction methods or reactions leading to obtaining morphine [7,8].

Drug reactions may be extracellular or cellular. The former involve noncellular constituents (physical effects, chemical reactions, physicochemical mechanisms, and modification of the composition of body fluids) [9]. In contrast, cellular mechanisms of action involve functional constituents of the cell and, more frequently, depend on specific biochemical reactions. Cellular drug reactions apply to physicochemical and biophysical mechanisms, the modification of cell membrane structure and function, and enzyme inhibition, as well as interactions with receptors. Receptors in various locations—for example, on the plasma membrane, in the cytosol, or in the nucleus—and interactions with the receptor typically activate or inhibit several sequences of biochemical events. The existence of receptors has not always been a matter of course in the world of science. Their first exponents were John Langley with his theory of receptive substances in cells and Paul Ehrlich and his side-chain theory [10,11]. These theories evolved, and a one-drug one-receptor model was proposed, based on the assumption that a given drug interacts only with one receptor.

Hitherto, drugs with high target selectivity in the body, designed to affect a single biological entity to avoid adverse side effects, have been used and searched for [12]. It is the concept of a “single molecule, single target, and single drug” that has dominated the pharmaceutical market for the last several decades. In some disease entities, a drug’s effect on one target is insufficient to achieve the therapeutic effect. Therefore, combination therapy is also used, i.e., the simultaneous use of drugs with different body targets—for instance, in the treatment of hypertension [13]. Diseases with multifactorial pathogenesis or not fully understood pathogenesis are often treated with more than one drug, which increases the risk of adverse drug reactions. Therefore, more and more often, as in the case of Alzheimer’s disease, drugs with multiple biological targets are chosen in order to increase effectiveness, safety, and to facilitate administration—for example, galantamine, caproctamine, or memoquin [12].

Unlike the treatment with synthetic drugs, phytotherapy is based on herbal medicines, and their actions result from the combined mechanisms of compounds contained in the raw materials or their products [14]. Thus, the effect of a given herbal drug is the result of all the synergies or antagonisms between its constituents. 

*Cannabis sativa* L. is one of the most ancient plant species used by humans for many purposes. Besides medicinal use, it also serves as a fiber; food; and is an important raw material for the production of rope, textiles, and paper. Surprisingly, the first human use of *Cannabis* is reported to be 10,000 years ago, at the end of the Ice Age [15]. The most ancient Chinese Pharmacopoea (written in the first century before the current era), the “Shen Nung Pen Ts’ao Ching”, is the first historical evidence of the use of *Cannabis* in traditional medicine. It includes all the traditional remedies used and orally bequeathed for over two thousand years when it was emperor Shen Nung’s (2700 years before the current era) reign [16]. The *Cannabis* plant first arrived in Europe with Scythians or proto-Scythians moving from Central Asia about 3500 years ago [17]. Modern reports on medical marijuana date back to the nineteenth century, when the Irish doctor William Brooke O’Shaughnessy performed experiments on the pharmacological and toxic properties of *Cannabis* [18,19]. He suggested Indian hemp as a treatment for tetanus and other convulsive diseases. In 1851, for the first time, *Cannabis* was included in the third edition of the United States Pharmacopoeia (USP) with the use of *Cannabis* flowers as an analgesic, anticonvulsant, and hypnotic [20]. In the second half of the nineteenth century, there was a significant increase in the use of *Cannabis* in medicine and research on its phytochemistry and pharmacology. At the beginning of the twentieth century, as the recreational use of *Cannabis* grew, in 1937, the “Marijuana Tax Act” was introduced, and in 1941, *Cannabis* was removed from the twelfth edition of the U.S. Pharmacopeia by the American Medical Association [19]. In the 1960s, the use of recreational *Cannabis* increased throughout the Western world. The discovery of the chemical structure of Δ^9^-tetrahydrocannabinol (Δ^9^-THC), which was identified by Gaoni and Mechoulam, boosted multidirectional research on *Cannabis* [21].

The botanical definition of *Cannabis sativa* L. has been a goal of many studies due to its morphological and chemical differentiation. The currently accepted classification distinguishes two subspecies—namely, ssp. *sativa* and ssp. *indica.* Within each of them, there are two main varieties, cultivated and wild. The most important from a medicinal point of view is *C. sativa* ssp. *sativa* var. *sativa* (so-called *C. sativa*) and *C. sativa* ssp. *indica* var. *indica* (so-called *C. indica*). *C. sativa* ssp. *sativa* var. *spontanea* (the so-called *C. ruderalis*) is rarely found and much less exploited. The plant develops both male and female specimens. Dried inflorescences from female plants (*Cannabis flos*) are known as marijuana. It contains resin, which is a source of phytocannabinoids. Purified resin is known as hashish. So-called *C. indica* is usually grown for recreational use, whereas so-called *C. sativa* has recently gained much attention as a source of medical marijuana. 

The possibility of interactions between components of various *Cannabis* products is high due to the richness of the compounds. Hitherto, over 100 cannabinoids have been identified in the *Cannabis* sp., and the best known, famous, and responsible for its pharmacological activity are the psychoactive Δ^9^-tetrahydrocannabinol (THC) and cannabidiol (CBD) [22]. These cannabinoids are produced in the plant in acid form and require decarboxylation, which is caused mainly by a high temperature. There are also other cannabinoids in the plant, such as cannabigerol (CBG), cannabichromene (CBC), cannabidivarin (CBDV), and cannabinol (CBN). However, the *Cannabis* sp. comprises not only cannabinoids. The presence of over 600 compounds has been reported so far, with over 150 different terpenes—including monoterpenes, sesquiterpenes, and, also, di- and triterpenes, as well as sugars, steroids, fatty acids, non-cannabinoid phenols, flavonoids, phenylpropanoids, alkanes, and nitrogenous compounds [23,24].

The grounds for the conclusion that there are interactions between *Cannabis* plant components is that the *Cannabis flos* extract has a higher potency than the individual isolated cannabinoids, which is called the “entourage effect” [22]. As a result, a more efficient pharmacological effect can be obtained and a higher therapeutic efficacy among patients. Individual cannabinoids can interact synergistically with each other, which is called the “intra-entourage effect”. The enhanced therapeutic effect of cannabinoids through other plant secondary metabolites such as terpenoids is called the “inter-entourage effect”. Terpenoids significantly alter the biological activity of cannabinoids.

*Cannabis* potency was initially defined by the content of THC. Over the years, the THC content in products available worldwide has elevated [25]. It is worth emphasizing that the use of products with high THC contents and low CBD contents may lead to side effects. A case-control study showed that the administration of *Cannabis* rich in THC might be associated with a raised risk of psychosis in patients, mostly when the CBD level was relatively low [26]. The THC/CBD ratio is significant in determining the potency of the *Cannabis* plant. CBD does not cause euphoria. Scientists even confirmed its antipsychotic and anxiolytic effect and the reduction of some side effects of THC [27]. Using CBD and THC together is considered to have a milder effect. Additionally, pretreatment with CBD before the administration of THC seems to be beneficial. In one study, it prevented the transient psychotic symptoms induced by THC [28]. That is why it is so crucial to antagonize the psychotic and other side effects of THC by using preparations containing both THC and CBD.

The vast majority of CBD products are readily available to patients worldwide as food supplements or additives in food. CBD is administered (with decreasing bioavailability) as: vaping/smoking; suppositories (vaginal/rectal); sublingual/oral mucosal; nasal sprays; capsules; and topicals (lotions, creams, shampoos, or oils). There are various high-THC products on the market: hashish, marijuana, and oils. Several products are registered for medical purposes. These include medical marijuana (Bedrocan^®^), oil preparations based on marijuana extracts [5], highly purified extracts containing a specific ratio of THC and CBD (Sativex^®^), and synthetic (Marinol^®^ and Cesamet^®^) cannabinoids [29]. In order to differentiate between synthetic and natural cannabinoids, the latter are usually referred to as phytocannabinoids. Synthetic cannabinoids available on the market include Dronabinol (Marinol^®^), which is indicated in adults for the treatment of anorexia associated with weight loss in patients with Acquired Immune Deficiency Syndrome (AIDS), nausea and vomiting associated with cancer chemotherapy in patients who have failed to respond well to conventional antiemetic treatments, and Nabilone (Cesamet^®^), which is indicated for the treatment of nausea and vomiting associated with cancer chemotherapy in patients who have failed to respond adequately to conventional antiemetic treatments [30]. When using synthetic analogs in medicine, the positive interactions between the secondary metabolites present in *Cannabis* plants are ignored. The synthetic compound does not have the opportunity to interact with other compounds that the plant has to offer and gain a similar pharmacological effect as CBD has on THC (an increase of the safety of using plant-based products). Maybe that is why the synthetic cannabinoid type 1 (CB_1_) receptor antagonist Rimonabant (Acompilia^®^) (used to reduce appetite in overweight and obese patients) was suspended from the market in 2008 and withdrawn from Europe in 2009 due to serious side effects, such as depression and anxiety disorders and several cases of suicide [17,29,31].

## 2. Endocannabinoid System

The endocannabinoid system is responsible for the regulation of numerous vital functions, including learning, memory, mood, anxiety, drug addiction, nutritional behavior, pain perception, modulation, and the functions of the cardiovascular system. It consists of cannabinoid receptors, endogenous ligands, secondary messengers, and endocannabinoid degradation pathways [32,33].

This system is still under investigation, but so far, three receptors have been discovered and described: CB_1_, cannabinoid type 2 (CB_2_), and G protein-coupled receptor 55 (GPR_55_). They are located extracellularly and intracellularly in the mitochondria, Golgi apparatus, and the nucleus [32]. The receptors that are located on cell membranes are coupled with the G protein. The endocannabinoid system is regulated by endocannabinoids, which are produced in response to neural network activity [34]. These are endogenous ligands produced by the human body. The two main ones are anandamide (AEA) and 2-arachidonylglycerol (2-AG) [32]. Their formulas are shown in Figure 1. They are derivatives of arachidonic acid, a precursor of numerous endogenous particles, such as prostaglandins, prostacyclins, thromboxanes, and leukotrienes.

These endocannabinoids are produced on demand from membrane precursors and phospholipids and are not stored in vesicles like other neurotransmitters but released immediately. Once made, they return to the presynaptic neurons to attach to receptors there. After their activation, the influx of potassium ions into the cell increases, leading to membrane hyperpolarization. As a result, the likelihood of neurotransmitters being released from presynaptic neurons is reduced. It makes it impossible to connect with other neurotransmitters, such as γ-aminobutyric acid (GABA), glutamate, or acetylcholine, affecting the central nervous system’s various functions. The stimulation of cannabinoid receptors may lead to a weakening of their responses, depending on the type of neurons they act on. After inducing a biological response, their reuptake by endocannabinoid membrane transporters takes place, followed by enzymatic degradation [33,34,35,36,37].

CB_1_ receptors mediate pain modulation, memory processing, motor function, and psychoactivity. They are located mainly in the brain. They achieve the highest density in such structures as the hippocampus, cerebellum, basal ganglia, cerebral cortex, hypothalamus, dorsal vagus nerve complex, and spinal cord. Moreover, several receptors are located in the structures responsible for the control of respiratory functions. The CB_1_ receptor predominates in the GABA inhibitory interneurons in the spinal cord’s posterior horn over excitatory neurons, with AEA being its major ligand. These presynaptic receptors influence the release of neuropeptides and neurotransmitters and inhibit synaptic conduction. Their activation stimulates potassium channels, which inhibit the presynaptic stimulation of neurons and block potentially dependent calcium channels, which reduce the release of neurotransmitters. CB_2_ receptors are mainly associated with the immune system and are mostly found in the spleen, digestive system, peripheral nervous system, and testes. Although several of them are located in the brain, their function in this organ is unclear. Knowing the receptors’ exact locations allows an understanding of how the system works and *Cannabis*’s effects on the body. The stimulation of this receptor does not have a psychoactive effect, nor does it affect the circulation. Its main endogenous ligand is 2-AG. The stimulation of these receptors reduces the release of chemokines and neutrophils and the migration of macrophages, and what is more, it has the effect of reducing chronic inflammatory processes and chronic pain. Both CB_1_ and CB_2_ found on mast cells are involved in the anti-inflammatory activity of cannabinoids. On the other hand, activation of the CB_2_ receptor on keratinocytes additionally increases the release of β-endorphins. The CB_2_ receptor stimulation process leads to the inhibition of adenylate cyclase activity, decreased cyclic adenosine monophosphate (cAMP) production, and increased activity of the mitogen-activated kinase. However, the effects on the calcium and potassium channels have not been confirmed [33,34].

Cannabinoid receptor agonists, such as THC, excite cannabinoid receptors nonselectively, producing a nonphysiological response, because hydrolytic enzymes do not metabolize them. Thus, their effect lasts longer and cannot be controlled. The long-term effects of THC may reduce the amount of the CB_1_ receptor and, consequently, weaken the sensitivity of the endocannabinoid system. The developed tolerance and decreased effect may lead to a gradual increase in consumption frequency or an increase in the dose. However, it is worth noting that this change is reversible just four weeks after discontinuation of use. Synthetic cannabinoids are considered “superagonists”, as they cause long-lasting effects on the endocannabinoid system, which can cause severe toxic effects [34]. Anandamide is broken down by fatty acid amide hydrolase (FAAH) into arachidonic acid and ethanolamine. The inhibition of FAAH increases the levels of AEA and the fatty acid amides oleoyl ethanolamide and palmitoylethanolamide. These compounds are no longer endocannabinoids, as they do not act on cannabinoid receptors. Still, they have anti-inflammatory activity and inhibit pain by acting on the transient potential type 1 vanilloid receptor (TRPV1) and binding to the peroxisome proliferator-activated receptor (PPAR). They can potentiate the effects of AEA by competitively inhibiting FAAH, which breaks down anandamide. AEA is a partial agonist of cannabinoid receptors; a full agonist of the TRPV1 receptor activates GPR_55_ and PPAR receptors and inhibits serotonin receptors (5-HT_3A_) and T-type potent calcium channels. 2-Arachidonylglycerol is a full agonist of the CB_1_ and CB_2_ receptors, and its amount exceeds the amount of AEA. It is broken down by monoacylglycerol lipase into arachidonic acid and glycerol [33,35,38]. CBD can lead to an increase in endocannabinoid levels, possibly altering their metabolism and regulating their reuptake. Therefore, the two above-mentioned endocannabinoids can be described as secondary messengers, formed in the process of phospholipid metabolism, playing a significant role in neuromodulation by preventing excessive neuronal activation [36,38].

According to many studies, the main functions of the endocannabinoid system include neuromodulation, stimulation of the central nervous system via CB_1_ receptors, anti-inflammatory and immunomodulating effects obtained by stimulation of the CB_2_ receptor, and neuromodulation by presynaptic inhibition [38]. The endocannabinoid system is supposed to maintain homeostasis in the body, and disturbances in its functioning may lead to various pathological states [39]. Some studies have shown an increase in the expression of cannabinoid receptors in neoplastic cells compared to healthy cells. In vitro studies on healthy human skin cells have shown that the activation of cannabinoid receptors initiates apoptosis in tumor cells and inhibits tumor angiogenesis [40].

## 3. Chemical Composition

As mentioned earlier, the *Cannabis* sp. contains numerous phytochemicals like phytocannabinoids, as well as alkanes; nitrogen compounds; terpenes; and phenols, including flavonoids, steroids, fatty acids, and sugars [41].

### 3.1. Phytocannabinoids

From a phytochemical and pharmacological point of view, phytocannabinoids appear to be the most crucial compounds [41]. They are produced mainly in *Cannabis* sp., although their presence has also been reported among plants belonging to the *Helichrysum* and *Radula* genus [42]. So far, over 100 different phytocannabinoids have been described in *Cannabis sativa* L., some of them being the breakdown products of original compounds [43]. Phytocannabinoids have 21 carbon atoms or, in the case of carboxylated forms, 22 [42]. Most often, they are tricyclic terpenoid compounds with a benzopyrene group in their structure. Based on the resorcinol side chain, phytocannabinoids have been classified into two classes: alkyl and β-aralkyl [43]. ElSohly M. A., on the other hand, grouped cannabinoids into 11 chemical classes, which are presented in the diagram below (Figure 2) [44]. The structures of the cannabinoids are shown in Figure 3.

Two pathways are involved in the production of cannabinoid precursors (Figure 4). The first is the polyketide pathway, which produces olivetolic acid. The second is the pathway located in plastids, leading to geranyl diphosphate (GPP). Initially, the hexanoate, which is a short-chain fatty acid, produces hexanoyl coenzyme A (CoA). The latter, in turn, by combining with three malonyl-CoA molecules, gives olivetolic acid, obtained thanks to the action of polyketide cyclase [42]. Then, under the influence of geranyl transferase, cannabigerolic acid (CBGA) is formed, which is the precursor for most cannabinoids [45]. Further transformations to the proper cannabinoids take place thanks to the appropriate oxidocyclases [18]. CBD and THC are synthesized by the independent cyclization of cannabigerolic acid [46]. The oxidase reaction to obtain CBD may be completed at this stage but may also arise during smoking or due to the acidic environment, together with small amounts of ∆^9^-tetrahydrocannabinol (∆^9^-THC) and ∆^8^-tetrahydrocannabinol (∆^8^-THC) [43]. However, this reaction does not occur naturally in the plant [46].

Quantitative data indicate that Δ^9^-tetrahydrocannabinolic acid (THCA), cannabidiolic acid (CBDA), and cannabinoic acid (CBNA) are the predominant primary phytocannabinoids of fresh *C. sativa* L. [42]. On heating, THCA and CBDA are converted into the two most essential phytocannabinoids produced by *C. sativa* L.—namely, Δ^9^-tetrahydrocannabinol and cannabidiol [47]. The *Cannabis sativa* ssp. *sativa* var. *sativa* (so-called *C. sativa*) usually contains more CBD than THC, while the *Cannabis sativa* ssp. *indica* var. *indica* (so-called *C. indica*) usually has more THC than CBD [48]. The *Cannabis sativa* ssp. *sativa* var. *spontanea* (the so-called *C. ruderalis*), in turn, is characterized by small levels of THC [49]. However, there are many hybrid varieties, such as Skunk (75% *sativa* and 25% *indica*) [29]. The hybridization process allows for the creation of new strains with selected characteristics. After creating hybrids with *C. ruderalis*, which has auto-flowering properties (flowers appear according to plant age rather than light conditions), the resulting hybrid has the appropriate cannabinoid content and auto-flowering properties [49]. High THC content strains are also obtained [49]. The European Drugs Report 2006–2014 showed a significant increase in THC’s average amount in resin and dried hemp. An analysis by the US Drug Enforcement Administration also confirms this trend. The average THC levels have risen from 4 to 12% over the past two decades. A similar phenomenon has been documented in Australia and the Netherlands. This is likely due to advanced production techniques and the desire to elevate THC levels. However, it should be borne in mind that it may also lead to increased toxicity [29]. 

The synthesis and accumulation of phytocannabinoids occur in glandular trichomes (epidermal outgrowths, consisting of the head and stalk), and their maximum concentration is noted in unfertilized female inflorescences [45]. Besides, they can also be found in the leaves and stem, and low levels of phytocannabinoids have also been detected in *Cannabis* seeds. Their contents were remarkably low in the seeds’ interiors and slightly higher on the surfaces, possibly due to contamination by flowers or leaves. The concentration of phytocannabinoids depends on the age, variety, type of tissue, environmental conditions during growth, storage conditions, and harvest time [42]. It has been found that phytocannabinoids are produced and stored mainly in the form of acids; therefore, their concentrations in fresh plants are much higher [41,43,45]. Neutral bioactive phytocannabinoids are formed by nonenzymatic decarboxylation during drying, storage, and heating as a result of the actions of temperature, light, and oxygen. 

Phytocannabinoids are soluble in lipids and nonpolar organic solvents [43]. Their lipophilicity is as follows: CBG > CBC > cannabitriol (CBT) > CBD > cannabielsoin (CBE) > THC > CBDV > CBN > cannabicycline (CBL) [50]. Their lipophilic nature means that they cross the blood–brain barrier and are readily distributed throughout the brain and nerve cell membranes. Moreover, they may gradually be released from the above structures into the bloodstream, even over many weeks [38].

#### 3.1.1. Cannabidiol-Type Compounds

Cannabidiol was first isolated in 1940, while its structure was described in the 1960s [51]. It is the main phytocannabinoid component of hemp. It does not have a psychoactive effect; it does not affect the locomotor activity, body temperature, memory, or sedation in the user and does not cause abuse but may have antipsychotic effects [29,51]. 

Some research groups have reported a lack of affinity of CBD for the CB_1_ and CB_2_ cannabinoid receptors. However, a weak antagonism of these receptors has been confirmed in vitro. CBD also inhibits anandamide reuptake and can increase anandamide levels according to the inhibition of FAAH. CBD also is an allosteric modulator of μ and δ opioid receptors [51]. In addition to the endocannabinoid system, it shows activity against serotonin receptors. Its anti-inflammatory and immunosuppressive effects may arise from the ability to stimulate A_1A_ adenosine receptors [52]. The exact influence of CBD on the receptors is described in Table 1. CBD can also reduce THC’s side effects, giving it a better safety profile. In its presence, at low concentrations, it can antagonize CB_1_, preventing tachycardia and anxiety and reducing sedation and hunger [53]. 

CBD has anti-inflammatory, antioxidant, analgesic, anxiolytic, antitumor activity, anticonvulsant, antiemetic, immunomodulatory, and neuroprotective effects [52,56]. The antiepileptic effects of CBD have been studied in the treatment of some types of epilepsy. The results showed possible efficacy in the treatment of childhood epilepsy. In June 2018, the oral solution of Epidiolex^®^ was approved by the US Food and Drug Administration. It has been approved for epileptic seizures in Lennox-Gastaut syndrome (LGS) and Dravet syndrome (DS) from the age of two [29]. The anxiolytic properties of CBD have been shown in both humans and animals. It is also under consideration for use in cancer, inflammation, diabetes, and neurodegenerative diseases. It has no harmful effects. CBD, being a component of nabiximols, has been used in the treatment of cancer pain in opioid nonresponders [53].

#### 3.1.2. Compounds of the Δ^9^-Tetrahydrocannabinol Type

The Δ^9^-tetrahydrocannabinol type includes many reduced cannabinol forms that differ from each other in a stereochemical configuration or double-bond position. The most important and best-known compound in this group is, however, Δ^9^-THC, which is the main component of *Cannabis* [43,46]. Its structure is shown in Figure 3. The first isolation of Δ^9^-THC took place in 1942 [57]. Only over twenty years later, Messrs. Raphael Mechoulam and Yechiel Gaoni described its structure based on a Lebanese hashish sample using magnetic resonance spectroscopy [21]. The determined localization of the double bond had never been considered in studies before. Šantavy defined the absolute configuration of the narcotic components, evidencing the relationship between the structure of Δ^9^-THC, Δ^8^-THC, and CBD [58,59]. The natural form of Δ^9^-THC in plants is a mixture of its acidic forms—THCA-A and THCA-B. They occur in plants in various concentrations, their decarboxylation takes place at different temperatures, and additionally, they have different crystalline forms. As a pure substance, Δ^9^-THC is unstable. It exists in an amorphous gum form and tends to brown. It shows greater stability in the unprocessed plant product, and it might be stored in a chilled methanol solution. The degradation rate of ∆^9^-THC in the plant product kept at room temperature is about 5%, while, for the pure substance, the value is around 10%. One of its degradation products is cannabinol [46].

As a partial agonist of CB_1_ and CB_2_ receptors, ∆^9^-THC adopts an agonist or antagonist profile, depending on the receptor expression, cell type, and the presence of full agonists or endocannabinoids. It is the compound responsible for the pharmacological effects of *Cannabis*. Δ^9^-THC also impacts other receptors. It is a GPR_55_ and GPR_18_ receptor agonist. It has anticancer activity, and it causes vascular relaxation, according to PPAR-γ receptor agonism. Moreover, its analgesic properties might result from the influence on glycine receptors. Some studies have proven the agonism of Δ^9^-THC against TRPV2, TRPV3, and TRPV4 channels, but no response to TRPV1 was observed [52]. The detailed activity of Δ^9^-THC on individual receptors is included in Table 2.

Even in early studies, the narcotic properties of *Cannabis* sp. were linked to the reduced form of cannabinol—the first structurally known cannabinoid [46]. Δ^9^-tetrahydrocannabinol is known for the ability to induce psychotropic effects. Its chronic administration causes various central and peripheral effects, confirmed in humans and animal models [41]. It exhibits anti-inflammatory, antiemetic, and neuroprotective properties, and it seems to be effective in the fight against chronic and neuropathic pain [60]. THC is also responsible for the addictive effect of *Cannabis* [29]. However, its acidic counterpart has antibiotic and neuroprotective potential [41,43]. Interestingly, compared to other cannabinoids, Δ^9^-THC has low acute toxicity at the median lethal dose (LD_50_) level of 100 mg/kg [46].

#### 3.1.3. Compounds of the Δ^8^-Tetrahydrocannabinol Type

Δ^8^-tetrahydrocannabinol might be formed according to CBD cyclization or a Δ^9^-THC molecule double-bond location change [61]. The CBD cyclization process varies depending on the reaction conditions. The Δ^9^ isomer is formed under mildly acidic conditions, while the Δ^8^ isomer is formed at a higher acidity and temperature. Δ^8^-THC is more stable than Δ^9^-THC, and it is easier to obtain, which may stimulate the isomerization towards Δ^8^-THC. Δ^8^-THC shows a similar impact on Δ^9^-THC as the receptors but with a slightly lower potency [46]. The structure of Δ^8^-THC is shown in Figure 3.

#### 3.1.4. Compounds of the Cannabigerol Type

The cannabigerol-type cannabinoids constitute the most structurally diverse class of phytocannabinoids. They can have various modifications of the isoprenyl group, the resorcinol core, and different substituents [46]. By 2005, eight compounds of this type had been reported, while nine new ones were added recently [62]. In this class of cannabinoids, CBG and cannabigerovarin (CBGV) are of the most significant importance [46].

The first compound isolated from *Cannabis* resin was cannabigerol [62]. The plant is rich in CBG, which has no psychoactive effects [43,52]. Its acid form, i.e., CBGA, may serve as a potential precursor for other cannabinoids [52]. Recently, a *Cannabis sativa* hybrid was obtained rich in this component [46]. The structure of cannabigerol in an acid or alkali environment is not stable [46]. Its structure is shown in Figure 3.

CBG inhibits anandamide reuptake, and the affinity for CB_1_ and CB_2_ receptors is low. It is a weak agonist of TRPV1 and TRPV2, a strong agonist of TRPA1, and a potent antagonist of TRPM8. Its non-cannabinoid activity is based on the ability to block the 5-HT_1A_ receptor and activate the α2 adrenergic receptor. CBG stimulates the α2 receptor; inhibits catecholamine release; and causes sedation, analgesia, and muscle relaxation [46]. Studies have demonstrated the possible neuroprotective effects of cannabigerol in Huntington’s disease [63]. Moreover, it might be a promising compound in the treatment of glaucoma, prostate cancer, and inflammatory bowel diseases. However, these applications require additional research [64]. Other sources report its antibiotic, antifungal, analgesic, antioxidant, and anti-inflammatory effects [65]. Unfortunately, the knowledge about cannabigerol, despite several studies conducted so far, remains limited for the time being [66].

#### 3.1.5. Compounds of the Cannabichrome Type

Claussen and, independently, Gaoni and Mechoulam revealed a cannabichromene structure in 1966 [67,68]. *Cannabis sativa* is rich in CBC. Its increased content in the plant is often associated with high THC levels, which may be related to a common biosynthetic pathway [52]. The CBC structure is presented in Figure 3. A high concentration of the acid form of CBC—cannabichromene acid—was found in young *Cannabis* plants, and its decrease progressed with the maturation of the plant [42]. It is a racemic compound that glows blue fluorescence under UV light and shows considerable stability [43,46]. It inhibits anandamide uptake. Cannabichromene is an agonist of the TRPA1 channels; it might block TPRM8 and activate the TRPV3 and TRPV4 channels. Its affinity for cannabinoid receptors is low, it is a selective CB_2_ receptor agonist, and it can also recruit CB_2_ receptor regulatory mechanisms [52,69]. The antibiotic, antifungal, anti-inflammatory, and analgesic effects of cannabichromene are noticed [69].

#### 3.1.6. Compounds of the Cannabinol Type

Cannabinol was the first phytocannabinoid isolated from *Cannabis* sp. Then, Wood, Spivey, and Easterfield were able to extract CBN from a sample of *Cannabis* originating in India in 1896 [70]. In 1940, Adams described its structure, and for the next two decades, it was the only compound described structurally in this class [71]. It is believed that derivatives, analogs, and CBN itself derive from THC-type compounds’ oxidative aromatization reaction. The conversion process from THC to CBN can occur spontaneously in both *Cannabis* extracts and raw plant materials. CBN is a stable compound. It has been used as a marker for identifying *Cannabis* in various archaeological finds [46]. It has a low affinity for the CB_1_ and CB_2_ cannabinoid receptors, but it is slightly higher for CB_2_ [43]. It is also an antagonist of the TPRM8 channels and a potent agonist of TRPA1. It has been described as a weak psychoactive compound [52]. Studies have indicated the antibiotic, sedative, anti-inflammatory, and anticonvulsant effects of CBN [65]. Its structure is shown in Figure 3.

#### 3.1.7. Compounds of the of Cannabitriol Type

Cannabitriol was first isolated in 1966 by Obata and Ishikawa [72]. It was obtained by the oxidation of Δ^9^-THC catalyzed by antibodies. However, its chemical structure was known ten years later. The content of CBT in the plant is low [43]. Its pharmacological properties are not yet known [72]. Its structure is shown in Figure 3.

#### 3.1.8. Compounds of the Cannabielsoin Type

The plant produces cannabielsoin in small amounts. It is found only in some varieties of *Cannabis sativa* L. It is a metabolite of CBD, which is obtained after photooxidation or biotransformation with tissue culture [43,52]. Elsa Boyanova was the first to isolate the first compound from this group in the laboratories of Raphael Mechoulam [46,73]. CBE is the main product in the cannabidiol pyrolysis, which is why it is likely to be found in the smoke generated during smoking *Cannabis* [46]. Its effect on cannabinoid receptors is unknown yet, but it is proven that cytochrome P450 1A1 (CYP1A1) is the molecular target of CBE [52,74]. The CBE pattern is shown in Figure 3.

#### 3.1.9. Compounds of the Cannabicyclol Type

Cannabicycline was initially named THC-III, according to its structural similarity to THC [46]. CBL is formed by heating cannabichromene [46]. This compound does not induce narcotic effects [46]. It has little affinity for the CB_1_ and CB_2_ receptors [43]. CBL inhibits acetylcholinesterase (AChE) less than THC or CBN but more than CBD or CBG; it is less lipophilic than most cannabinoids [50]. The CBL structure is shown in Figure 3.

#### 3.1.10. Compounds of the Cannabinodiol Type

Cannabinodiol (CBND) was first described in 1977 [75]. It is an analog of cannabidiol obtained by its photochemical conversion; its content in the plant is low [52]. Its pharmacological properties have not been described yet. However, it is known that CBND shows psychoactive effects [52]. The CBND structure is presented in Figure 3.

### 3.2. Terpenes

Another biologically active group of compounds found in *Cannabis sativa* L. are terpenes. There are over 150 different terpenes in *Cannabis*. Most of them are not unique for *Cannabis* sp.; they occur in many plants from various families. Terpenes are synthesized like cannabinoids in trichomes. The mean contents of terpenes in *C. sativa* L. vary from 0.5–3.5%; they also can be identified in smoke and fumes generated during smoking or vaporization [60]. There seems to be a close relationship between the plant’s cannabinoid and terpene contents [45]. Being the main secondary metabolites besides cannabinoids, they could be used as chemotaxonomic markers [42], but they were not taken into account in the classification systems used so far [76]. Terpenes are classified according to the number of five-carbon fragments known as isoprene units. Those found in *Cannabis* include ten-carbon monoterpenes, fifteen-carbon sesquiterpenes, and thirty-carbon triterpenes (Figure 5).

The mono- and sesquiterpenes shown in Figure 5 have been found in parts of the plant, such as the leaves, roots, and flowers. Among monoterpenes, we can distinguish the predominant constituents of the volatile terpene fraction: D-limonene, linalool, β-myrcene, terpinolene, and α- and β-pinene. Sesquiterpenes (Figure 5), including β-caryophyllene and α-humulene, are found in significant amounts in *Cannabis* extracts. Triterpenes (Figure 5) are found in *Cannabis* roots. These are friedelin and epifriedelanol, as well as β-amyrin in *Cannabis* fibers. Cycloartenol, dammaradienol, and β-amyrin were detected in hempseed oil [42].

There are two main biosynthetic pathways for plant-derived terpenes, as shown in Figure 6. The first is the plastidic methylerythritol phosphate (MEP) pathway responsible for the production of monoterpenes, diterpenes, and tetraterpenes. The second, the cytosolic mevalonic acid (MVA) pathway, is involved in the synthesis of sesquiterpenes and triterpenes. The initial substrates of the pathways are pyruvate and acetyl-CoA. They are converted to isopentenyl diphosphate (IPP) and then isomerized to dimethylallyl diphosphate (DMAPP). Within the MEP pathway, the condensation of IPP and DMAPP occurs and the formation of GPP, which is a precursor to monoterpenes. In the MVA pathway, two IPP molecules and one DMAPP are combined, resulting in the formation of farnesyl diphosphate (FPP), which is sesquiterpenes’ and triterpenes’ precursor [42].

Studies have been conducted comparing the terpene profile of *Cannabis sativa* ssp. *indica* var. *indica* (so-called *C. indica*) and *Cannabis* ssp. *sativa* var. *sativa* (so-called *C. sativa*), in order to distinguish them according to their terpene profiles. In plants of the former variety, higher amounts of β-myrcene and limonene or α-pinene were found. On the other hand, the terpene profiles of the *Cannabis sativa* ssp. *sativa* var. *sativa* (called *C. sativa*) proved to be more complex. Some plants were more abundant in α-terpinolene or α-pinene, while, in others, β-myrcene, accompanied by α-terpinolene or trans-β-ocimene, was predominant [77].

The different terpene profiles have an impact on the odor and the taste of Cannabis [60,76]. The smell of *Cannabis sativa* ssp. *sativa* var. *sativa* (so-called *C. sativa*) is described as herbal or sweet, while the aroma of *Cannabis sativa* ssp. *indica* var. *indica* (so-called *C. indica*) is defined as irritating and smelly [48]. According to the fact that terpene compounds are the source of the plant’s characteristic smell, some of them, such as β-caryophyllene and caryophyllene oxide, are used to train dogs to enable the detection of marijuana or hashish [41].

Some terpenes are found only in the fresh plant due to their volatile nature—for example, α-pinene and limonene [41]. The environmental conditions and plant maturity influence the amount of terpenes produced by the plant and its distribution.

Terpenes found in *Cannabis* have a wide variety of biological properties [76]. Nerolidol, constituting 0.09% of *Cannabis* terpenes, is active against malaria and leishmaniasis [42]. D-limonene has cytotoxic in vitro, immunostimulatory, and anxiolytic properties. β-amyrin has been recognized as a compound with analgesic, anti-inflammatory, and anxiolytic properties. α-Pinene shows an improvement in memory capacity as it acts antagonistically to memory deficits caused by Δ^9^-tetrahydrocannabinol, inhibiting acetylcholinesterase. *Cannabis* contains linalool, also present in *Lavandula angustifolia,* showing anxiolytic, analgesic, anticonvulsant, and anti-inflammatory effects. β-Caryophyllene, also found in copaiba balm and black pepper, has anti-inflammatory and cytoprotective properties for stomach cells. It also binds to the CB_2_ receptor. Pentacyclic triterpenes, such as β-amyrin and cycloartenol, show antibacterial, antifungal, antitumor, and anti-inflammatory properties [42].

Triterpenes might play an essential role in *Cannabis* biological activity and change the properties of the cannabinoids [60]. Products containing cannabinoids, such as Sativex^®^, Cannador^®^, and Bedrocan^®^, may contain terpenes as support. As mentioned above, due to the “entourage effect” hypothesis, cannabinoids are more pharmacologically effective with terpenes than themselves [78]. Terpenes may alter the pharmacokinetics of THC by increasing the permeability of the blood–brain barrier [79]. Moreover, they may also affect their affinity to CB_1_ receptors and influence the analgesic and psychotic effects of cannabinoids related to their interaction with neurotransmitter receptors [80]. Thus, the effects of *Cannabis* are likely to be the sum of the interactions between cannabinoids, terpenoids, and flavonoids [79]. However, no clinical trials confirming these effects have been conducted [78]. Combinations of cannabinoids and terpenes are already used to treat certain conditions. Some of them are presented in Table 3.

### 3.3. Phenolic Compounds

Phenolics, including flavonoids, lignans, and stilbenes, are yet another group of compounds found in *Cannabis* [42]. Phenolic compounds are synthesized from phenylalanine through the cytoplasmic phenylpropanoid pathway. This precursor is phenylalanine, which can be transformed into esters, lignans, lignans, flavonoids, stilbenes, and coumarins. These compounds are formed mostly from tyramine and esters of coenzyme A with coumaric acid (p-coumaroyl-CoA), caffeic acid (caffeoyl-CoA), and coniferic acid (coniferyl-CoA). The course of the synthesis of flavonoids is shown in the diagram (Figure 7) [42].

Twenty-six flavonoids, which mainly represent flavonols and flavones, were isolated from *Cannabis* sp., including apigenin, luteolin, kaempferol, vitexin, isovitexin, orientin, and quercetin [80,81]. There is a considerable variation in the content of flavonoids between different varieties of *C. sativa* L., as well as between the individual parts of this plant. Their presence has been confirmed in flowers, twigs, leaves, and pollen. Methylated isoprenoid flavones, cannflavin A and cannflavin B, first isolated in 1980 by Crombie and Jamieson were also reported (Figure 8) [82]. The biological activity of cannflavins seems to be connected with their structure: a prenyl side chain—a geranyl (C10—cannflavin A) or a dimethylallyl (C5—cannflavin B) group, which is attached to the six position of the flavone A-ring—and modification at the 3′ flavone B-ring position with a methoxy group, which elevates the lipophilicity and results in a higher permeability through biological membranes [83,84,85]. In 2008, another compound from this rare group—cannflavin C—was isolated [81]. It seems that the germination process elevates the production of the mentioned cannflavins [80].

*Cannabis* sp. flavonoids have antibacterial, anti-inflammatory, antiproliferative, and neuroprotective properties. Flavonoids can affect the pharmacokinetics of Δ^9^-tetrahydrocannabinol [80]. Besides, apigenin has estrogen and anxiolytic properties. Cannflavins A and B, due to the inhibition of 5-lipoxygenase and prostaglandin E_2_, show anti-inflammatory effects [42]. Moreover, cannflavin B exhibits antimicrobial and antileishmanial activity [80].

Two groups of lignans were detected in *C. sativa* L.: lignanamides and phenolic amides. They occur in the roots, fruits, and seeds of the plant [81]. The lignanamides found in *Cannabis* include cannabisins and grossamide. For the first time, cannabisin A was detected from *Cannabis sativa* in 1991 [86]. Later, more and more cannabisins were identified. Cannabisins M, N, and O were identified quite recently, in 2015 [87]. They have antioxidant, anti-inflammatory, and neuroprotective properties [87,88]. Cannabisin D is present in *Cannabis* leaves, and its amount increases strongly with exposure to UV light. Significant levels of lignans such as syringaresinol, pinoresinol, lariciresinol, and secoisolariciresinol were determined in the hydrophilic extract obtained from *Cannabis* seeds. However, it was found that the total amount of lignans in *Cannabis* seeds is about 20 times lower than in flax seeds and is only 32 mg per 100 g of dry weight [89]. Moreover, the content in the seed shell is only 1% of the lignans content in the whole seed. Lignans have antiviral, antioxidant, antidiabetic, and anticancer properties and can be used in the treatment of obesity [90]. Additionally, some of the lignans mentioned earlier can transform, under the influence of anaerobic intestinal microflora, into enterolignans, which are the precursors of estrogens for mammals. Due to their similar structure, enterolignans and estrogens may potentially be used in the future in the treatment of certain hormone-dependent cancers. In vitro studies have described the cytoprotective and anti-inflammatory properties of the lignanamides [42].

Nineteen stilbenes containing phenanthrene, bibenzyl, and spiran rings in their structures were isolated from *Cannabis* plants. Among them were cannabistilbene I, IIa, and IIb and dihydroresveratrol. Under the influence of UV light, the number of dibenzyl stilbenes in *Cannabis* leaves was elevated [42]. Canniprene, belonging to the dihydrostilbenoids, inhibits the activity of proinflammatory eicosanoids [80]. Tiantian Guo et al. isolated the following stilbenoids from *Cannabis sativa* L. leaves.: α,α′-dihydro-3,4′,5-trihydroxy-4,5′-diisopentenylstilbene, combretastatin B-2, α,α′-dihydro-3′,4,5′-trihydroxy-4′-methoxy-3-isopentenylstilbene, α,α′-dihydro-3,4′,5-trihydroxy-4-methoxy-2,6-diisopentenylstilbene, and α,α′-dihydro-3′,4,5′-trihydroxy-4′-methoxy-2′,3-diisopentenylstilbene and cannabinoids [91]. The isolated compounds were cytotoxic to human cancer cells by inhibiting their proliferation and inducing cell death. Moreover, stilbenoids can increase the cholesterol transfer to hepatocytes by improving the protein expression and acid bile synthesis.

### 3.4. Fatty Aids

The *Cannabis* sp. contains also fatty acids that can be found mainly in hempseeds. Hempseeds also include proteins (20–25%), carbohydrates (20–30%), oil (25–35%), and insoluble fiber (10–15%) and minerals [92,93]. The fatty acid composition of *Cannabis sativa* seeds was investigated in the last century. The presence of both saturated and unsaturated acids was determined [94]. The following acids were identified: arachidic, linoleic, behenic, eicosadienoic, lignoceric, palmitic, eicosenoic, linolenic, sativic, stearic, myristic, oleic, palmitoleic, and cisvaccenic from different *Cannabis* seed preparations [94]. The different varieties of *Cannabis* may differ in both the qualitative and quantitative compositions of fatty acids. However, there is no correlation between the seeds’ fatty acid composition and their geographical origin [94]. As the seeds mature, the composition of the fatty acids changes—the more mature the seed, the higher the percentage of unsaturated fatty acids and the lower the percentage of saturated [94]. Unsaturated fatty acids are essential for cell structure. They modulate cell membrane fluidity, structural maintenance, and have a role in cell signaling [95]. Fatty acids are precursors for the biosynthesis of many functional metabolites [95]. Fatty acids also have antioxidant properties and regulate the nervous system, blood pressure, hematic clotting, inflammatory processes, and glucose tolerance [96]. *Cannabis* seeds have a desirable ratio of omega-6 and omega-3 polyunsaturated fatty acids considered optimal for human nutrition. The consumption of the fatty acids in the ratio correlates with the reduced occurrence of degenerative cardiovascular disease, depression, and neurodegenerative diseases like Alzheimer’s disease [97,98].

## 4. Pharmacological Application

The effectiveness of extracts or drugs from *Cannabis* sp. or synthetic cannabinoids in diseases (Figure 9.) has been shown below. Data indicating the clinical effectiveness of cannabinoid compounds in such a wide array of disease units is an argument for their multitarget action. The ability to antagonize or agonize CB_1_ and CB_2_ receptors results in the stimulation or inhibition of the endocannabinoid system, both in the peripheral tissues and in the central nervous system. As indicated below, *Cannabis* sp. extracts have significant therapeutic potential. Still, due to the need to continue further clinical trials, they should be used as one of the last therapeutic procedures.

### 4.1. Epilepsy

Epilepsy is believed to be the most common noninfectious neurological disease. Moreover, one-third of patients suffer from seizures resistant to standard antiepileptic drugs. Antiepileptic drugs have been used for over 160 years. They prevent seizures, but there is no evidence that they have disease-modifying properties. They affect the mechanisms causing the brain’s excitability. Antiepileptic drugs disrupt the creation and spread of epileptic hyperactivity. These drugs act on the following mechanisms: (i) modulation of voltage-dependent sodium channels (carbamazepine and phenytoin), calcium channels (ethosuximide), and potassium channels (retigabine and ezogabine); (ii) enhancement of inhibited GABA transmission (benzodiazepines, tiagabine, and vigabatrin); (iii) weakening of the stimulant glutamate transmission (perampanel); and (iv) presynaptic modification of neurotransmitter release (levetiracetam, brivaracetam, gabapentin, and pregabalin) [99]. Drugs introduced before 1989 are called “first-generation” drugs, and those introduced later are named “second-generation” drugs. Medicines that have been recently approved (brivaracetam, eslicarbazepine acetate, lacosamide, perampanel, rufinamide, and stiripentol) that are improvements to classic drugs or rely on novel mechanisms are called “third-generation” drugs [100]. Since 2018, the last group has also included CBD.

Drug-resistant epilepsy is diagnosed when symptom control cannot be achieved despite the use of at least two different antiepileptic drugs in an appropriate dose [101]. Epilepsy with the greatest resistance to treatment includes LGS, DS, fever-related epilepsy syndrome, and epilepsy in tuberous sclerosis syndrome (TSC) [102]. The patient’s quality of life and cognitive and psychosocial problems significantly deteriorate.

Due to the occurrence of drug-resistant epilepsy and severe side effects associated with the use of traditional antiepileptic drugs, preparations with higher safety and effectiveness are being sought. The antiepileptic use of *Cannabis* dates back to ancient times. It is generally accepted that *Cannabis* benefits in anticonvulsant therapy are linked to CBD. However, the mechanism of CBD action in the treatment of epilepsy is not fully understood. The orphan G-protein coupled receptor, GPR_55_ antagonism, is mentioned as the mechanism. Recent studies have shown that CBD significantly reduces the number of seizures. Moreover, it seems that only mild side effects occur. Among these, increased liver enzymes levels are improved with a continuity of therapy or by dose reduction. However, it should be emphasized that CBD is most often used in studies with standard antiepileptic therapy, so it is difficult to assess whether it has antiepileptic effects or only enhances the effects of traditional drugs [103]. The FDA approved a 99% CBD extract called Epidiolex in 2018 for the treatment of seizures associated with LGS, DS, or TSC in patients one years of age and older [104].

In recent years, there has been an increased interest in the use of *Cannabis* as part of the treatment of childhood epilepsy, among others, due to numerous media reports on children whose symptoms improved with *Cannabis*. Many parents of children struggling with drug-resistant epilepsy turn to alternative therapeutic measures, sometimes with doctors’ help and sometimes on their own [101]. A lot of attention was focused on a Colorado family who was looking for alternative treatments for a girl suffering from DS, Charlotte Figi. Her supportive treatment included a CBD-rich *Cannabis* strain that was later named “Web Charlotte.” After three months of using it, the number of seizures dropped from 300 per week to 30 per month. However, after 20 months of treatment, the monthly number of attacks was about three, and the girl started talking and walking [104].

The effectiveness of CBD against the drug-resistant seizures that occur in DS has been studied. DS is a type of childhood epilepsy with a low efficacy of traditional antiepileptic drugs and a high mortality rate. A randomized, double-blind clinical trial with a placebo control was performed. The groups were randomly selected among 120 children and young adults, and the patients were subjected to standard antiepileptic treatment. Participants were administered 20-mg/kg body weight CBD daily as an oral solution or placebo. The results were reported after 14 weeks of treatment and are presented in Table 4 [105]. CBD is more effective than a placebo in reducing epileptic seizures in patients suffering from DS. At the same time, such therapy carries a greater risk of side effects [105].

In a phase 3 clinical trial, 225 LGS patients aged 2–55 years with two or more seizures per week were administered CBD oral solution—10 mg/kg body weight per day or twice, in addition to a conventional antiepileptic regimen to patients in the test group and placebo for 14 weeks. The median percent reduction in seizure frequency from the baseline was 37.2% in the CBD 10 mg/kg/day group, 41.9% in the CBD 20 mg/kg/day group, and 17.2% in the placebo group [106]. The reduction in seizure frequency was dose-related.

Among 171 patients aged 2–55 years diagnosed with LGS, whose seizure frequency over four weeks was at least two per week and who did not respond to treatment with at least two antiepileptic drugs (the most common were lamotrigine, valproate, and clobazam), 86 patients received CBD at a dose of 20 mg/kg/day for 14 weeks; the rest obtained a placebo [107]. After 14 weeks of treatment, the monthly seizure frequency had decreased by a median of 43.9% from the baseline in the CBD group. A 50% or more reduction in seizure frequency was reported in 44% of patients in the CBD group and 24% of the placebo group. Serious adverse events (pneumonia, viral infection, increase in alanine aminotransferase, increase in aspartate aminotransferase, and increase in γ-glutamyltransferase) occurred in 23.26% of patients in the CBD-treated group and in 4.71% in the placebo group. Serious treatment-emergent (increased levels of alanine aminotransferase, aspartate aminotransferase, and γ-glutamyltransferase) occurred in four patients in the CBD group. Instead, the most common not serious adverse events (vomiting, diarrhea, loss of appetite, and somnolence) occurred in 61.63% of the CBD group and 50.59% of the patients in the placebo group. The study showed many side effects in patients using CBD, but in this group, the reduction in seizure frequency was greater than in the placebo group.

### 4.2. Sclerosis Multiplex

Multiple sclerosis is the most common inflammatory demyelinating disease of the central nervous system. It arises as a result of an autoimmune response directed against myelin, which, in consequence, leads to worsening neurodegeneration and disability [108]. Common symptoms of multiple sclerosis include vision problems, pain, muscle weakness, balance problems, and paralysis [109]. In addition, patients who suffer from spasticity, which is defined as the activation of muscles without the patient’s will, are under increased risk of complications and are subject to pain and problems with falling asleep [110]. These symptoms are caused by progressive abnormal neuronal transmission. The disease is initially characterized by long asymptomatic periods abruptly interrupted by relapses, usually lasting several days or weeks [109]. The beginning of modern treatment of multiple sclerosis can be associated with the introduction of interferon beta and glatiramer acetate in the treatment of multiple sclerosis with relapsing remission [111]. Then, monoclonal antibodies were introduced, such as natalizumab, and, later, also alemtuzumab and ocrelizumab. Oral medicines such as fingolimod, teriflunomide, dimethyl fumarate, and cladribine have also been developed.

New therapeutic options are still being sought, especially for continuously progressive forms of the disease. Currently, the available therapies may delay disease progression and reduce the number of attacks but do not improve the patients’ quality of life [56,108].

Numerous clinical and preclinical studies give hope that cannabinoids derived from *Cannabis sativa* L. can be used to control chronic pain and spasticity. Preclinical studies have also demonstrated the potential use of endocannabinoids’ neuroprotective properties and partial inhibition of disease progression [108]. Nabiximols, THC, and oral *Cannabis* extract are believed to be helpful in the treatment of spasticity. Sativex^®^ is currently the only pharmaceutical product approved for this indication. However, Sativex^®^ is not a first-line treatment. It is indicated for the relief of moderate-to-severe spasticity symptoms in adult sclerosis multiplex (SM) patients who are unresponsive to other spasticity-relieving medicinal products and show a clinically significant improvement in spasticity symptoms in the initial stage of therapy. Medical marijuana reduces the pain and spasticity in SM but negatively affects balance and posture. Nabiximols may also be effective in disturbance of the normal bladder function, which is common in this condition [109]. The effects of individual cannabinoids on pain and muscle spasticity in SM patients are shown in Table 5.

Studies of the use of medical marijuana in the treatment of SM suggest a reduction in spasticity after using phytocannabinoids, but it is not statistically significant. However, there was a clinically significant subjective reduction in spasticity reported by patients. Cannabinoids appear to be well-tolerated by patients using them as add-on therapy. It enables them to improve their score on the Ashworth Scale, which is a measure of spasticity by measuring the resistance that occurs when soft tissue is passively stretched. However, many studies give contradictory results, often without statistical significance. Unfortunately, the use of *Cannabis* harms cognitive functions, and as many as 40–60% of SM patients suffer from it, so when starting cannabinoid therapy, the patient’s cognitive functions should be monitored. Medical marijuana has a positive impact on SM symptoms. However, medical marijuana does not slow the disease’s progression [56].

### 4.3. Vomiting and Nausea Prevention

Approximately 45–61% of cancer patients suffer from chemotherapy-induced nausea and vomiting (CINV). In antiemetic prophylaxis during chemotherapy, the following are used: (i) drugs from the group of 5-HT_3_ receptor antagonists—ondansetron, granisetron, tropisetron, dolasetron, palonosetron; (ii) drugs from the group of neurokinin-1 receptor antagonists—aprepitant; (iii) corticosteroids—the most commonly recommended dexamethasone or methylprednisolone; and (iv) complementary drugs: dopamine receptor antagonists (metoclopramide), phenothiazine derivatives (chlorpromazine), antihistamines, and butyrophenone derivatives (haloperidol and droperidol) [112]. The guidelines propose the use of antiemetic drugs in monotherapy or according to a schedule of two-to-three drugs, depending on the degree of risk of nausea and vomiting and the individual sensitivity of the patient. If the antiemetic prophylaxis proves ineffective, additional antiemetic drugs are used when needed. An acute episode due to the stimulation of gastrointestinal receptors occurs up to 24 h after the initiation of chemotherapy, and delayed CINV occurs within one–five days, mainly due to the activation of receptors located in the brain. Statistically, the delayed effect is more frequent, and the incidence of nausea outweighs the incidence of vomiting [113].

According to studies, the incidence of nausea and vomiting can be reduced by activating the CB_1_ receptor with THC [113]. As a result of receptor stimulation, the proemetic effects of dopamine and serotonin are abolished [114]. In the fight against CINV, oral cannabinoid preparations are more effective than a placebo. However, compared with other antiemetics, studies lead to different conclusions. Some of them suggest the superiority of THC, others, a similar effectiveness, and still others show that the best effect is obtained when combining traditional drugs with cannabinoids [110].

After reviewing randomized controlled trials, the conclusions are that administering cannabinoids to patients during chemotherapy allows for 70% control of nausea and 66% control of vomiting, while, after administration of a placebo, the results reached 57% and 36%, respectively [115]. During the research carried out in the seventies and eighties, THC’s superiority in the prevention of nausea and vomiting occurring during radio- and chemotherapy over the placebo and antiemetic drugs used at that time was found. Currently, however, we are already in possession of more effective drugs for this indication. The comparisons show that newer drugs can control nausea in 90% of patients, compared to 30% with cannabinoids. For this reason, they are not used in the first line of treatment but only as an adjunct or second-line treatment [116]. Cesamet^®^ or Marinol^®^ can be used in the treatment of CINV [56].

### 4.4. Pain

The endocannabinoid system is involved in pain control, and therefore, *Cannabis* may be used to relieve it [56]. The most common medical use of *Cannabis* is pain management, but it is not equally effective for all types of pain. Most likely, it results from different mechanisms of pain formation. Studies show that cannabinoids cannot combat acute pain, but they can only relieve chronic pain to a limited extent [33]. The pharmacotherapy of chronic pain is incredibly difficult, as is the pain that continues despite the tissue’s healing.

Neuropathic pain is a type of chronic pain that occurs following damage to the peripheral or central nervous system. It appears as a result of an injury, disease, cancer, immune disorder, or an effect of certain drugs. Often, patients then experience allodynia, hyperalgesia, depression, sleep problems, and anxiety, and their contact with society is usually limited. According to Special Interest Group on Neuropathic Pain, the guidelines for the treatment of neuropathic pain include: (i) first-line therapy: gabapentinoids—gabapentin and pregabalin, tricyclic antidepressants—amitriptyline, and serotonin-norepinephrine reuptake inhibitors—duloxetine and venlafaxine; (ii) second-line therapy: opioids—tramadol and tapentadol and topical treatment—lidocaine and capsaicin; and (iii) third-line therapy: strong opioids—morphine and oxycodone and a neurotoxin—botulinum toxin [117]. Due to the side effects of using available drugs in the treatment of neuropathic pain, and the fact that more than 50% of people do not achieve the expected improvement, new therapeutic agents are sought continuously [118].

### 4.5. Increasing Appetite

Anorexia may appear in the course of such chronic diseases as AIDS, cancer, or anorexia nervosa [56]. Progestational agents (megestrol) are often used to increase the appetite, and in the short-term, the use of corticosteroids, the oral ghrelin mimetic anamorelin, can be used [119]. However, there are many limitations to their use. Hormonal manipulation may be contraindicated in many patients, and long-term corticosteroid therapy may lead to metabolic changes, more fractures, cataracts, gastrointestinal discomfort, and changes in mood or behavior. Seven patients with hematogenous metastatic melanoma and liver metastases suffering from extensive loss of appetite and nausea were administered dronabinol (synthetic delta-tetrahydrocannabinol) capsules for four weeks [120]. Most of the patients reported increased appetite and decreased nausea. These results lasted for several weeks. A more extensive, randomized study showed that oral Dronabinol could help stimulate appetite in advanced cancer patients and slow weight loss. The study involved 469 people who had lost weight due to advanced cancer. They were divided into three groups. The first received 5 mg of Dronabinol daily, the second 800 mg of megestrol daily, and the third received both drugs. In the group of people receiving only megestrol, their appetite was increased by 75%, while their weight by 11%. In those receiving only Dronabinol, these values were 49% and 3%, respectively, and these differences were statistically significant. On the other hand, the administration of both drugs did not provide additional benefits. Numerous animal studies have also confirmed an increase in appetite and the amount of food consumed after the administration of cannabinoids. Studies in the 1970s showed that smoking marijuana increases the number of calories you eat. It seems crucial that, in people affected by cancer, medical marijuana will have antiemetic and appetite-increasing effects [31]. Marinol^®^ has been approved as a drug increasing appetites in patients with AIDS and cancer [116]. 

Cannabinoids have been established as possible for use in the third line of neuropathic pain treatment, right after antidepressants, anticonvulsants, and opioids. Health Canada has also implemented guidelines recommending *Cannabis* smoking to combat noncancerous chronic pain. Ultimately, it was established that there is substantial evidence of a positive effect of *Cannabis* use in chronic pain patients [33]. The best results in the treatment of neuropathic pain are achieved with low doses of THC/CBD or high doses of CBD [118].

### 4.6. Inflammatory Bowel Diseases

Inflammatory bowel diseases include ulcerative colitis and Crohn’s disease. Although the etiology of these diseases has not been elucidated so far, it is assumed that directing the human immune system against its own intestinal microflora contributes to their formation. In Crohn’s disease, conventional medication includes aminosalicylates, corticosteroids, thiopurines, methotrexate, and thalidomide. Antitumor necrosis factors (infliximab, adalimumab, and certolizumab) and other biologics such as antibodies (vedolizumab and ustekinumab) are also used [121].

The endocannabinoid system is probably responsible for maintaining intestinal homeostasis, because when disturbances occur, it triggers the synthesis of effector molecules so that it may be involved in intestinal inflammation. The presence of the endocannabinoid system within the entire gastrointestinal tract has also been demonstrated. *Cannabis* has been used in enteritis since ancient times, but unfortunately, research into its use in this indication is rare. Survey studies have shown that patients use *Cannabis* to self-medicate to relieve inflammatory bowel disease symptoms such as abdominal pain, loss of appetite, and diarrhea. Unfortunately, most studies are characterized by insufficient statistical significance and a problem with the selection of an appropriate placebo, as the effects of *Cannabis* on the central nervous system are difficult to mask [122].

In an anonymous survey, IBD patients reported that *Cannabis* relieved abdominal pain (83.9%), abdominal cramps (76.8%), arthralgia (48.2%), and, to a lesser extent, diarrhea (28.6%) [123]. A prospective, placebo-controlled study was conducted to test the ability of *Cannabis* to remission Crohn’s disease. The study involved 21 patients who did not respond to standard treatment, and their Crohn’s disease activity index (CDAI) was over 200. In the study group, patients received a cigarette containing 115 mg of THC twice a day, while, in the second group, patients received a placebo in the form of THC-free *Cannabis* flowers. Their conditions were assessed during the eight-week treatment and, also, two weeks after its completion. Forty-five percent (5/11) of the test group achieved complete remission (CDAI decrease <150), while, in the placebo group, this effect was found in 10% (1/10) of people. In contrast, a decrease in the CDAI score by a minimum of 100 was observed in 10/11 people receiving *Cannabis* and 4/10 in the placebo group. It is also worth noting the lack of reported adverse effects of the applied treatment [124].

### 4.7. Parkinson’s Disease

Parkinson’s disease is a neurological condition involving the degeneration of nigrostriatal dopamine neurons. The disease’s symptoms are primarily muscle stiffness, tremor at rest, bradykinesia, and lack of postural stability. Additionally, depression, anxiety, orthostatic hypotension, constipation, fatigue, and sleep problems may appear. Modern pharmacotherapy consists of supplementing the deficiency of dopamine and symptomatic treatment. In patients with Parkinson’s disease, treatment is symptomatic and focuses on improving the motor and nonmotor symptoms [125]. No disease-modifying drugs are available. Drugs that affect the motor impairment include levodopa, dopamine agonists (ropinirole and pramipexole), monoamine oxidase inhibitors (selegiline and rasagiline), catechol-O-methyltransferase inhibitors (entacapone), anticholinergic drugs (trihexyphenidyl), and choline inhibitors. Depending on the symptoms, antipsychotics or antidepressants are also used. According to this procedure, positive effects on the motor symptoms are obtained, but also, side effects, such as dyskinesia, appear. For this reason, a solution that would holistically approach the treatment of this disease is continuously being sought. Components of the endocannabinoid system are involved in regulating motor functions and dopamine activity. Therefore, *Cannabis* seems to be an attractive therapeutic option in the treatment of movement and neurodegenerative disorders [126]. Studies have found medical marijuana to help combat Parkinson’s disease’s motor symptoms, such as stiffness, tremors, and bradykinesia. In addition, it provides pain relief and reduces sleep disturbance in patients [127].

### 4.8. Tourette’s Syndrome

Tourette’s syndrome’s main symptoms are numerous motor tics and the presence of at least one vocal tic. This disease affects approximately 1% of the population, and men are affected by the disease three-to-four times more often than women. The therapy of Tourette’s syndrome may consist of neuropsychological interventions; pharmacotherapy: (i) drugs that block or lessen dopamine—haloperidol and risperidone, (ii) botulinum injections, (iii) attention-deficit hyperactivity disorder medications—methylphenidate and dextroamphetamine, (iv) central adrenergic inhibitors—clonidine and guanfacine, (v) antidepressants—fluoxetine, and (vi) antiseizure medications: topiramate; and deep brain stimulation [128].

The treatments available today often fail to give the desired results or cause serious side effects, and many patients are looking for alternatives or additive supplements to their standard therapy to relieve their symptoms [129]. They reach for dietary supplements or stimulants such as alcohol, nicotine, or marijuana, among others. Two patients using Sativex^®^ showed a clear improvement in motor and vocal tics [56]. However, there is still very limited evidence for THC’s efficacy in relieving the symptoms of Tourette’s syndrome, only from small clinical trials [130].

### 4.9. Schizophrenia

Schizophrenia is a mental illness emerging most frequently in early adulthood. It involves so-called positive and negative symptoms and cognitive dysfunction. The positive symptoms include hallucinations, paranoia, and delusions, while the negative symptoms include emotional withdrawal, lack of motivation, and decreased affect. The main drugs in the treatment of schizophrenia are antipsychotics (neuroleptics): (i) first generation—haloperidol, perazine, and chlorpromazine; (ii) second generation—olanzapine, quetiapine, and risperidone; and (iii) third generation—aripiprazole, brexpiprazole, and cariprazine. Complementary, non-neuroleptic drugs are also used—antidepressants, sedatives, hypnotics, and mood stabilizers [131].

Studies have shown an increased use of marijuana in people with schizophrenia from the general population. It has been proven that these patients’ endocannabinoid systems are disrupted. The number of their CB_1_ receptors is increased. Besides, consuming *Cannabis* sp. elevates the risk of developing schizophrenia. Their harmfulness, however, depends on the content of THC and CBD, because THC promotes psychosis while CBD has an antipsychotic effect. The long-term use of *Cannabis* disrupts neuronal synchronization in a similar way to people with schizophrenia. *Cannabis* adversely affects the positive symptoms, and its use leads to more frequent relapses of the disease after achieving remission. However, according to *Cannabis* use, an improvement in the negative symptoms was found [109].

### 4.10. Glaucoma

Glaucoma is an eye disorder characterized by optic neuropathy with visual field loss. It is often accompanied by increased pressure in the eyeball and results in the irreversible decrement of vision. Glaucoma can be treated with medication (eye drops), laser therapy, or surgery. The eye drops used: beta blockers—timolol, levobunolol, and carteolol; prostaglandins—latanoprost, travoprost, and tafluprost; alpha-adrenergic agonists—brimonidine and apraclonidine; carbonic anhydrase inhibitors—dorzolamide and brinzolamide; and cholinergic drugs—pilocarpine and carbachol [132].

Studies using oral, intravenous, and inhaled medical marijuana have shown that Δ^9^-THC reduces the intraocular pressure, which is a risk factor for glaucoma. The patients (60–65%) experience a 25% reduction in intraocular pressure [133]. Moreover, THC has been shown to have neuroprotective potential [134]. The intraocular pressure lowering effect is probably related to the reduction of water production. However, this condition does not last long, necessitating frequent administration of the drug. Another disadvantage is the blood pressure reduction of marijuana and its potential ability to reduce ocular perfusion [135]. There have also been studies using THC in the form of eye drops, but these were less effective. This approach would avoid systemic side effects but require the development of a more effective delivery system [133]. Unfortunately, the advantage of marijuana in terms of safety and effectiveness over currently available glaucoma treatments has not been demonstrated so far [110].

### 4.11. COVID-19

At the end of 2019, the first-identified cases of a pneumonia of unknown origin emerged from China. The National Health Commission of China reported on the epidemic in early 2020 [136]. Initially, the virus was called “novel coronavirus 2019” (2019-nCoV) by the World Health Organization (WHO), but the name was changed to “severe acute respiratory syndrome coronavirus 2” (SARS-CoV-2) by the international committee of the Coronavirus Study Group (CSG), while the disease was designated “coronavirus disease 2019” by the WHO [137]. This virus is very contagious and has rapidly spread globally. On 11 March 2020, the epidemic of COVID-19 disease caused by the SARS-CoV-2 coronavirus was declared a pandemic by the WHO. Until then, more than 118,000 people across 114 countries, territories, and areas were infected by this virus [138]. During treatment, already known drugs are being used. Neither a vaccine nor a specific cure has been developed yet [139].

The virus is transmitted directly from human-to-human (via coughing, sneezing, and the spread of respiratory droplets or aerosols) or indirectly (via contaminated surfaces). Therefore, isolation is a crucial element in slowing down the spread of the novel virus [138]. Patients go through infection with varying severity, from asymptomatic to mild symptoms through severe stages to fatal cases. According to estimates, approximately 70% of COVID-19 patients are asymptomatic or with mild disease, while the remaining 30% are severe cases [140]. The disease progression may depend on the patient’s genetic differences in the immune system and the individual’s exposome [141].

A mild course of illness symptoms are cough, fever, headache, shortness of breath, muscle pain, sore throat, diarrhea, vomiting, and the most characteristic loss of taste and/or smell [140]. In the severe disease, dyspnea appears, respiratory frequency is ≥30/min, and blood oxygen saturation (SpO_2_) is ≤93%. The percentage of oxygen supplied is (fraction of inspired oxygen, FiO_2_) < 300, and/or lung infiltrates are >50% within 24 to 48 h. Severe pneumonia, Acute Respiratory Distress Syndrome, as well as extrapulmonary symptoms and systemic complications such as sepsis and septic shock are observed in patients

In the case of critically ill patients, high concentrations of interleukins (ILs)—IL-2, IL-7, and IL-10; granulocyte colony-stimulating factor (G-SCF); monocyte chemoattractant protein-1 (MCP1/CCL2); macrophage inflammatory protein 1 alpha (MIP1-α/CCL3); CXC-chemokine ligand (CXCL10/IP10); tumor necrosis factor (TNF-α); C-reactive protein (CRP); ferritin; and D-dimers in SARS-CoV-2 are observed in patients’ plasma [142]. The disease’s high severity is related to the cytokine storm, which is possibly induced by the interleukin-6 amplifier [143]. Respiratory failure, septic shock, and/or dysfunction or failure of multiple organs appear in the critical disease [144].

The mechanism of SARS-CoV-2 attacking human cells is similar to that of SARS-CoV. Coronavirus proteins bind to angiotensin-converting enzyme 2 (ACE2) receptors, which allows them to introduce into the host’s cells viral RNA [145]. Extracts with a high content of CBD show the ability to downregulate enzymes: ACE2 and transmembrane serine protease 2 (TMPRSS2), which makes SARS-CoV-2 challenging to enter the human body through oral airways [146,147]. Therefore, it could be used as a prophylaxis before COVID-19. Cannabidiol exhibits direct antioxidant activity by influencing the level and activity of antioxidants and oxidants. It decreases reactive oxygen species (ROS) generation. Moreover, it affects the redox balance according to indirect mechanisms like the modulation of CBD, PPAR-γ, GPR_55_, and TRP channel receptors 5-HT_1A_ [148]. Moreover, CBD as a PPAR-γ agonist causes pulmonary inflammation limitation, a decrease of fibrosis, and the inhibition of viral replication and immunomodulation. CBD also has anti-inflammatory properties due to the inhibition of cytokine production: IL-1α and β, IL-2, IL-6, interferon-gamma, inducible protein-10, monocyte chemoattractant protein-1, macrophage inflammatory protein-1α, and TNF-α [149,150,151]. *Cannabis* constituents THC and CBD also inhibit T-helper type 1 (Th1) cytokines and/or promote an in vitro and in vivo Th2 immune response [152]. Th1 and the inflammatory immune response profile predominate in COVID-19. Some viruses may take advantage of host inflammation, and others may be detrimental to host inflammation [152]. In COVID-19, hyperinflammatory immune responses can have serious consequences, such as increased disease severity and patients’ mortality. Therefore, the immunomodulatory effects of cannabinoids can help reduce the severity of symptoms. However, it should be remembered that *Cannabis* is an unconventional immunomodulatory agent. Its route of administration through smoking is contraindicated in patients suffering from COVID-19. Therefore, inhalation (extracts) or oral administration should be considered to increase patient safety [152]. Unfortunately, there is no reliable evidence in humans confirming effective anti-inflammatory doses for CBD [153].

## 5. Conclusions

The complexity of the plant matrix represented by *Cannabis* sp. makes it possible to discover prototypes of many compounds that will show pharmacological activity within the endocannabinoid system or already show activity against receptors for which the participation and mechanisms of action in the stimulation of selected biochemical processes have been confirmed. To achieve the therapeutic goal, it is also essential to obtain the appropriate bioavailability of the compounds at the site of action. In the case of *Cannabis*, it is possible due to the synergism of phytocannabinoids and terpenes absorption. The various pharmacological actions for selected combinations of chemical compounds present in *Cannabis* presented in this review constitute an argument for clinical effectiveness based on a multi-fragment affinity for multiple receptors. Therefore, the necessity to search for new indications of the pharmacological application of *Cannabis* products is justified with regards to the search for target points within the endocannabinoid system and outside of it. However, it should be taken into account that the “entourage effect” of the compounds present in *Cannabis* raw materials will always be an important variable for the effectiveness and safety of therapy with their use. 

## Figures and Tables

**Figure 1 ijms-22-00778-f001:**
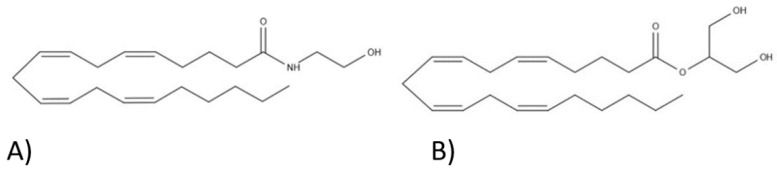
Endocannabinoid structures: (**A**) anandamide and (**B**) 2-arachidonoylglycerol.

**Figure 2 ijms-22-00778-f002:**
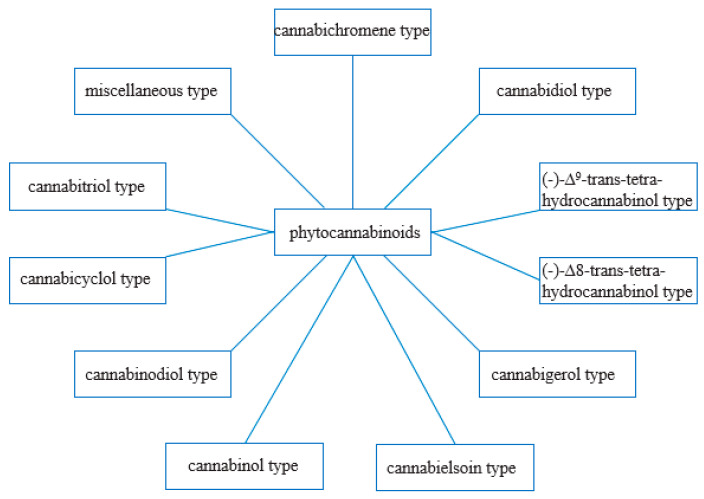
Classes of phytocannabinoids, according to ElSohly M. A. [44].

**Figure 3 ijms-22-00778-f003:**
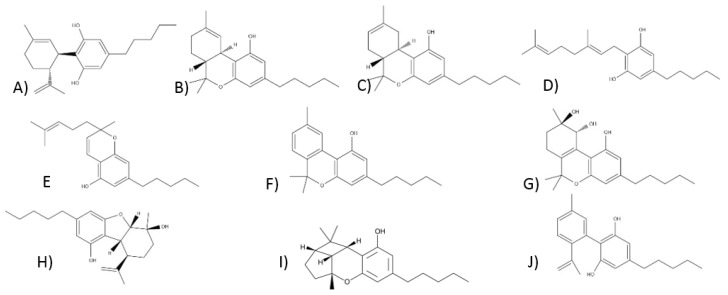
The structures of cannabinoids: (**A**) cannabidiol, (**B**) Δ^9^-tetrahydrocannabinol, (**C**) Δ^8^-tetrahydrocannabinol, (**D**) cannabigerol, (**E**) cannabichromen, (**F**) cannabinol, (**G**) cannabitriol, (**H**) cannabielsoin, (**I**) cannabicyclol, and (**J**) cannabinodiol.

**Figure 4 ijms-22-00778-f004:**
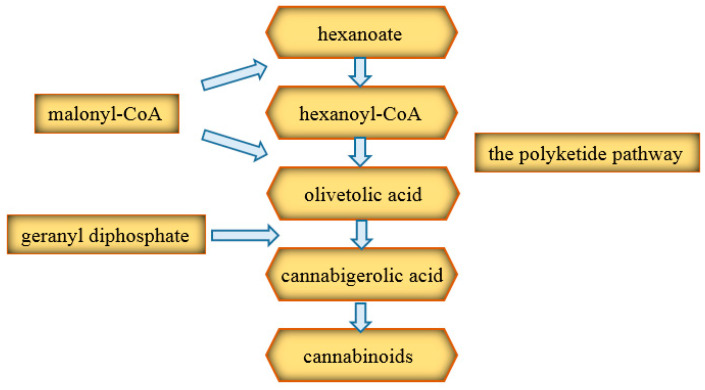
Diagram of cannabinoid biosynthesis [42]. CoA: coenzyme A.

**Figure 5 ijms-22-00778-f005:**
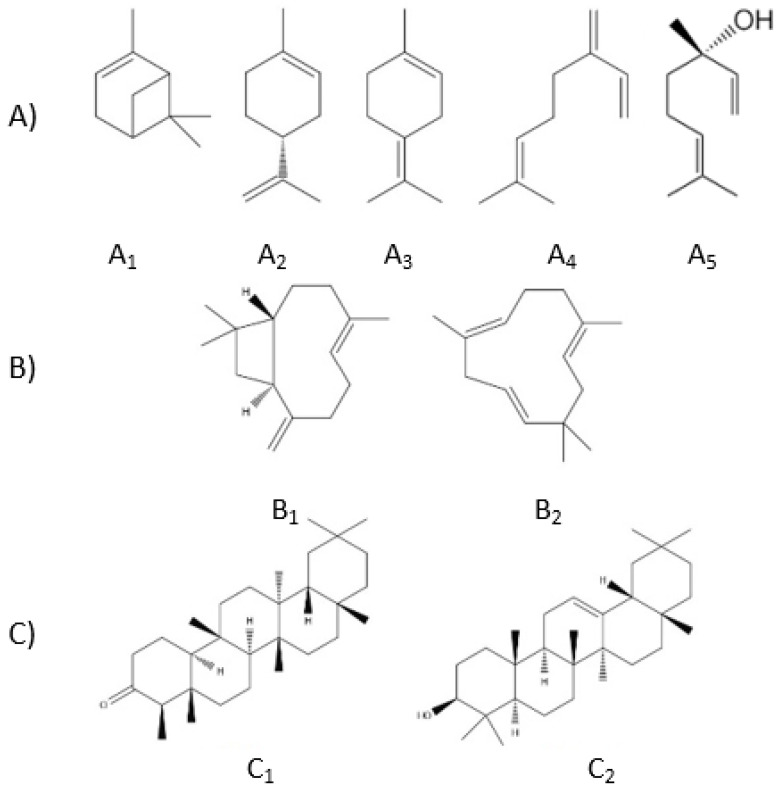
The structures of: (**A**)—monoterpenes (**A_1_**—α-pinene, **A_2_**—limonene, **A_3_**—terpinolene, **A_4_**—α-myrcene, and **A_5_**—linalool); (**B**)—sesquiterpenes (**B_1_**—β-caryophyllene and **B_2_**—α-humulene); and (**C**)—triterpenes (**C_1_**—friedelin and **C_2_**—β-amyrin).

**Figure 6 ijms-22-00778-f006:**
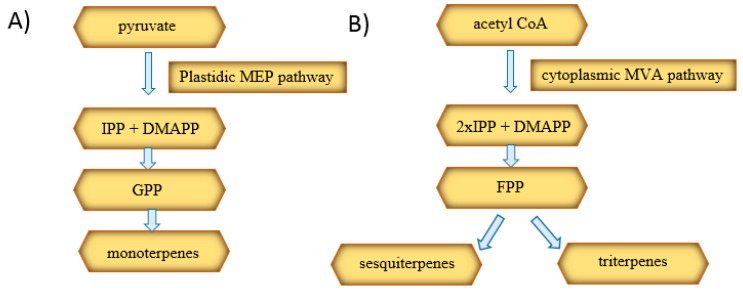
Diagram of the biosynthesis of monoterpenes (**A**), sesquiterpenes, and triterpenes (**B**) [42]. MEP: methylerythritol phosphate, IPP: isopentenyl diphosphate, DMAPP: dimethylallyl diphosphate, GPP: geranyl diphosphate, MVA: mevalonic acid, and FPP: farnesyl diphosphate.

**Figure 7 ijms-22-00778-f007:**
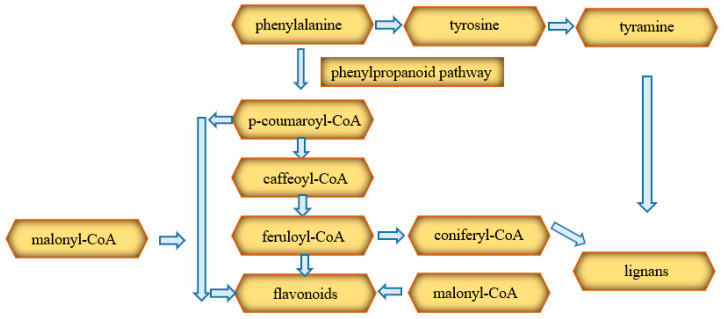
Diagram of the biosynthesis of phenolic compounds [42].

**Figure 8 ijms-22-00778-f008:**
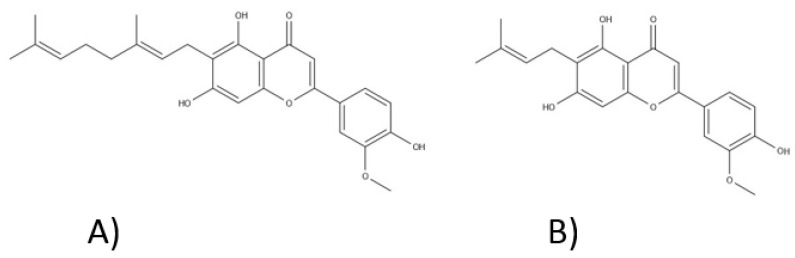
The structures of cannflavins: (**A**)—cannflavin A and (**B**)—Cannflavin B.

**Figure 9 ijms-22-00778-f009:**
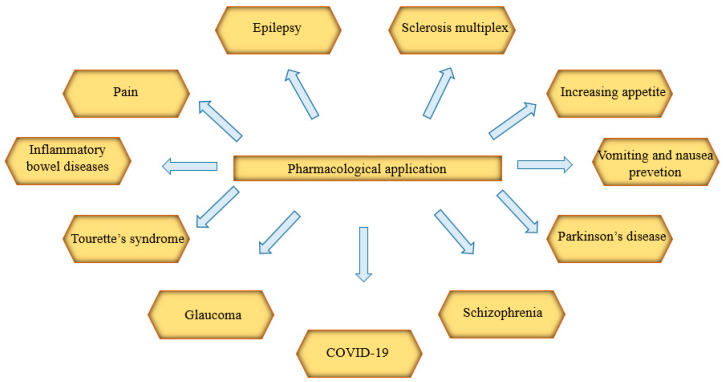
Pharmacological applications of extracts or drugs from *Cannabis* sp. or synthetic cannabinoids in diseases. COVID-19: coronavirus disease 2019.

**Table 1 ijms-22-00778-t001:** Postulated effects of cannabidiol (CBD) on the receptors [52,54,55]. Abbrev.: A—adenosine, TRPA1—transient receptor potential ankyrin 1, TRPV—transient potential vanilloid receptor, CB—cannabinoid, GPR—G protein-coupled receptor, and PPAR—peroxisome proliferator-activated receptor, 5-HT – serotonin receptor, TRPM – transient receptor potential melastatin.

Receptor	CBD Activity	Receptor	CBD Activity
CB_1_	Noncompetitive antagonist, negative allosteric modulator	5-HT_1A_	Agonist
CB_2_	Antagonist, inverse agonist	5-HT_2A_	Partial agonist
GPR_55_	Antagonist	5-HT_3A_	Antagonist
GPR_18_	Antagonist	A_1A_	Agonist
PPAR-γ	Agonist	channel TRPV1, TRPV2	Agonist
TRPM8	Antagonist	TRPA1	Antagonist
α1, α3 glycine receptors	Agonist		

**Table 2 ijms-22-00778-t002:** The effect of Δ^9^-tetrahydrocannabinol (Δ^9^-THC) on the receptors [52].

Receptor	Δ^9^-THC Activity	Receptor	Δ^9^-THC Activity
CB_1_	Partial agonist	opioid receptors	Allosteric modulator
CB_2_	Partial agonist	PPAR-γ	Agonist
GPR_55_	Agonist	TRPV2, TRPV3, TRPV4	Agonist
GPR_18_	Agonist	TRPA1	Agonist
5-HT_3A_	Antagonist	TRPM8	Antagonist

**Table 3 ijms-22-00778-t003:** Examples of the use of combinations of cannabinoids with terpenes [64]. CBG: cannabigerol.

Cannabinoids	Terpenes	Use
CBD	limonene, linalool, pinene	acne treatments
CBG	pinene	antiseptics
CBD	limonene, linalool	anxiety disorders
Extract CBD/THC 1:1	caryophyllene, linalool, myrcene	sleep disorders

**Table 4 ijms-22-00778-t004:** Results of the efficacy of CBD in the treatment of epileptic seizures in Dravet’s syndrome in children [105].

	Test Group	Control Group
Number of patients	61	59
Applied treatment	20 mg/kg body weight of cannabidiol daily	Placebo
Number of seizures per month after four weeks of standard treatment	12.4	14.9
Number of seizures after 14 weeks of study treatment	5.9	14.1
The proportion of patients in whom the number of seizures decreased by at least 50%	43%	27%
Percentage of patients who were completely seizure-free	5%	0%

**Table 5 ijms-22-00778-t005:** Cannabinoid therapies and their impacts on the symptoms of multiple sclerosis determined in clinical trials [109]. ↓: decrease.

The Form	Effects on Symptoms
smoked marijuana	spasticity ↓, pain ↓
nabiximols	spasticity ↓, pain ↓
oral *Cannabis* extract	muscle spasticity ↓
dronabinol	spasticity ↓, pain ↓
nabilone (+gabapentin)	pain ↓

## Data Availability

Data available in a publicly accessible repository (see list of references).

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
