# Peer review of "Cannabis sativa L. as a Natural Drug Meeting the Criteria of a Multitarget Approach to Treatment"

_ijms, 2021, doi:10.3390/ijms22020778_

Round 1

Reviewer 1 Report

Althought, the manuscript is written at a very high level, I have a few comments a mistakes in references:

line 290, at the begining of sentence: what does it mean "ElSohly"? Is it mistake? If it is reference, it shluld be than ref. 22 instead of ref. 27. Also same mistake in reference is on Figure 2.

line 299; Figure 3. At the figure l) the stereochemistry should be improved in formula of cannabicyclol. 

line 396; As a reference is listed 38, but there is no sign of name Mechoulam and Gaoni as author of listed publication. I recomend to author carreful check of all references.

line 479; Cannabis sp. (only one dot instead of two is sufficient).

line 480 also mistake in references. All the references in the text are shifted.

line 545; methylerythritol phoshate is highlighted

line 533; Figure 5 - stereochemistry of linalool might be indicated (β - OH, α-CH3)

line 535; instead of β-amarin(inccorect name) shloud be β-amyrine or IUPAC name - repeataed also line 543

line 550 dimethylallyl diphosphate is highlighted

Abbreviations:

line DMAPP and MEP are highlighted.

Please check all references carefully. Obviously, they have been shifted in several places and the references in the text do not correspond.

I recommend accepting the manuscript after a minor revision.

Author Response

Dear  Reviewer,Molecular Endocrinology and Metabolism, Endocannabinoid System in Health and Disease: Current Situation and Future Perspectives 3.0 We would like to kindly thank You for Your thorough review that helped us improve our paper. We have taken into account all the suggestions and have made the necessary changes. The changes in the uploaded manuscript are implemented using the "Track Changes" function in
Microsoft Word. However, because changes in the bibliography created by the special program are not seen in the "Track Changes" function, all of the changed reference numbers are highlighted in yellow. Our responses are as follows:

Comments from the Reviewer:

  1. line 290, at the begining of sentence: what does it mean "ElSohly"? Is it mistake? If it is reference, it shluld be than ref. 22 instead of ref. 27. Also same mistake in reference is on Figure 2.

Response: ElSohly stands for Mahmoud A. ElSohly, Ph.D. The authors apologize for such a mistake. The references have been thoroughly updated and in line 290 (now line 293) and in Figure 2. the work has been cited: ElSohly, M.A.; Slade, D. Chemical constituents of marijuana: the complex mixture of natural cannabinoids. Life Sci. 2005, 78, 539–548.

  1. line 299; Figure 3. At the figure l) the stereochemistry should be improved in formula of cannabicyclol. 

Response: The authors thank You very much for this comment, the stereochemistry in formula of cannabicyclol at the Figure 1. has been improved.

  1. line 396; As a reference is listed 38, but there is no sign of name Mechoulam and Gaoni as author of listed publication. I recomend to author carreful check of all references.

Response: The authors thank You very much for this comment, the reference was corrected to: Gaoni, Y.; Mechoulam, R. Isolation, structure, and partial synthesis of an active constituent of hashish. J. Am. Chem. Soc. 1964, 86, 1646–1647. We appreciate the suggestion to improve our manuscript. The whole list of the references was thoroughly checked and improved.

  1. line 479; Cannabis sp. (only one dot instead of two is sufficient).

Response: The authors apologize for such a mistake. Punctuation has been corrected.

  1. line 480 also mistake in references. All the references in the text are shifted.

Response: The authors thank You very much for this comment. In the line 480 (now 485) the reference was introduced: Wood, T.B.; Spivey, W.T.N.; Easterfield, T.H. III.—Cannabinol. Part I. J. Chem. Soc. Trans. 1899, 75, 20–36. All of the references in the manuscript were thoroughly checked and improved.

  1. line 545; methylerythritol phoshate is highlighted

Response: Highlighted words has been replaced with not highlighted words. The authors apologize for such a failure.

  1. line 533; Figure 5 - stereochemistry of linalool might be indicated (β - OH, α-CH3)

Response: The authors thank You very much for this comment, the stereochemistry for linalool has been added to the Figure 5.

  1. line 535; instead of β-amarin(inccorect name) shloud be β-amyrine or IUPAC name - repeataed also line 543

Response: The authors apologize for such a mistake. The name β-amarin was replaced with the name β-amyrin.

  1. line 550 dimethylallyl diphosphate is highlighted

Response: As requested by the Reviewer highlighted words were replaced with not highlighted words. The authors apologize for such a mistake.

  1. Abbreviations: line DMAPP and MEP are highlighted.

Response: The authors apologize for such a mistake. Appropriate corrections have been made in the abbreviations section.

  1. Please check all references carefully. Obviously, they have been shifted in several places and the references in the text do not correspond. I recommend accepting the manuscript after a minor revision.

Response: The authors really appreciate this comment.  We thank You for the suggestion to improve our manuscript. The list of references was carefully checked, all the mistakes were thoroughly implemented.

Reviewer 2 Report

Comments to Authors

In this review article entitled "Cannabis sativa L. as a natural drug meeting the criteria of a multitarget approach to treatment" by Anna Stasiłowicz et al, discuss about cannabis chemistry and medicinal uses in different pathological conditions. Subject content of review article is an excellent but the way review is presented in not convincing.

Several sentences are incomplete and difficult to follow the message. on certain critical places references are  missing. There is a missing connection between sentences.

Covering different pathological conditions in this review article is an excellent idea but presentation is of poor quality.

Addition of COVID section is not very helpful here, since this field is not well developed yet.

Chemistry section is the best part and that should be elaborated in more details.

Author Response

Dear  Reviewer,Molecular Endocrinology and Metabolism, Endocannabinoid System in Health and Disease: Current Situation and Future Perspectives 3.0 We would like to kindly thank You for Your thorough review that helped us improve our paper. We have taken into account all the suggestions and have made the necessary changes. The changes in the uploaded manuscript are implemented using the "Track Changes" function in
Microsoft Word. However, because changes in the bibliography created by the special program are not seen in the "Track Changes" function, all of the changed reference numbers are highlighted in yellow. Our responses are as follows:  

Comments from the Reviewer:

In this review article entitled "Cannabis sativa L. as a natural drug meeting the criteria of a multitarget approach to treatment" by Anna Stasiłowicz et al, discuss about cannabis chemistry and medicinal uses in different pathological conditions. Subject content of review article is an excellent but the way review is presented in not convincing.

  1. Several sentences are incomplete and difficult to follow the message. on certain critical places references are  missing. There is a missing connection between sentences.

Response: The authors thank You for these comments, perhaps these mistakes resulted from our inexperience in writing the review. All the comments are valuable to us. In the manuscript have been introduced linguistic changes, the order of the words in some sentences was changed, the cause-and-effect relationships on which we based our work were combined. The list of references was thoroughly checked. New references were added to the list.

  1. Covering different pathological conditions in this review article is an excellent idea but presentation is of poor quality.

Response: We appreciate the suggestion to improve our manuscript. We have added a graphic ordering all pharmacological applications in a clear manner. We hope that the quality of this section is now improved.

  1. Addition of COVID section is not very helpful here, since this field is not well developed yet.

Response: The authors appreciate this comment. Indeed, data on the use of Cannabis in COVID-19 as well as the topic of COVID-19 is fresh. There is not so much research containing Cannabis in this area. However, we believe that readability will be enhanced by showing such pioneering research. We are aware that these studies are at a different level than studies in other diseases, but we believe that it is worth emphasizing the latest reports. If the work is approved, we leave the decision to remove subsection 4.11. COVID-19 to the Editor.

  1. Chemistry section is the best part and that should be elaborated in more details.

Response: Thank you for the positive comment about the chemistry section being the best. It is very kind of the Reviewer. We believe that the chemistry section is the best according to our greatest experience in this area. We bear in mind that there is no comprehensive review of the chemistry of Cannabis components. As the Reviewer requested, we have also expanded this section.

Round 2

Reviewer 2 Report

No Comment required